# Single Seed-Based High-Throughput Genotyping and Rapid Generation Advancement for Accelerated Groundnut Genetics and Breeding Research

Sejal Parmar [1,2], Dnyaneshwar B. Deshmukh [1], Rakesh Kumar [3], Surendra S. Manohar [1], Pushpesh Joshi [1], Vinay Sharma [1], Sunil Chaudhari [1], Murali T. Variath [1], Sunil S. Gangurde [1,2], Rajaguru Bohar [1], Prashant Singam [2], Rajeev K. Varshney [1,4], Pasupuleti Janila [1,*] and Manish K. Pandey [1,*]

[1] International Crops Research Institute for the Semi-Arid Tropics (ICRISAT), Hyderabad 502324, India; P.Sejal@cgiar.org (S.P.); dnyanubdeshmukh@gmail.com (D.B.D.); m.surendra@cgiar.org (S.S.M.); j.pushpesh@cgiar.org (P.J.); s.vinay@cgiar.org (V.S.); sunil.chaudhari@worldveg.org (S.C.); v.murali@cgiar.org (M.T.V.); g.sunil@cgiar.org (S.S.G.); R.Bohar@cgiar.org (R.B.); r.k.varshney@cgiar.org (R.K.V.)

[2] Department of Genetics, Osmania University, Hyderabad 500007, India; prashantsingam@gmail.com

[3] Department of Life Sciences, Central University of Karnataka, Gulbarga 585367, India; rakeshkumar@cuk.ac.in

[4] State Agricultural Biotechnology Centre, Centre for Crop and Food Innovation, Food Futures Institute, Murdoch University, Murdoch, WA 6150, Australia

[*] Correspondence: p.janila@cgiar.org (P.J.); m.pandey@cgiar.org (M.K.P.)

**Abstract:** The groundnut breeding program at International Crops Research Institute for the Semi-Arid Tropics routinely performs marker-based early generation selection (MEGS) in thousands of segregating populations. The existing MEGS includes planting of segregating populations in fields or glasshouses, label tagging, and sample collection using leaf-punch from 20–25 day old plants followed by genotyping with 10 single nucleotide polymorphisms based early generation selection marker panels in a high throughput genotyping (HTPG) platform. The entire process is laborious, time consuming, and costly. Therefore, in order to save the time of the breeder and to reduce the cost during MEGS, we optimized a single seed chipping (SSC) process based MEGS protocol and deployed on large scale by genotyping >3000 samples from ongoing groundnut breeding program. In SSC-based MEGS, we used a small portion of cotyledon by slicing-off the posterior end of the single seed and transferred to the 96-deep well plate for DNA isolation and genotyping at HTPG platform. The chipped seeds were placed in 96-well seed-box in the same order of 96-well DNA sampling plate to enable tracking back to the selected individual seed. A high germination rate of 95–99% from the chipped seeds indicated that slicing of seeds from posterior end does not significantly affect germination percentage. In addition, we could successfully advance 3.5 generations in a year using a low-cost rapid generation turnover glass-house facility as compared to routine practice of two generations in field conditions. The integration of SSC based genotyping and rapid generation advancement (RGA) could significantly reduce the operational requirement of person-hours and expenses, and save a period of 6–8 months in groundnut genetics and breeding research.

**Keywords:** groundnut; seed-chipping; high-throughput genotyping; marker based early-generation selection; rapid generation advancement

## 1. Introduction

The selection of desirable plant type in the segregating population holds the key to rapid varietal improvement. The selection process has evolved from direct visible selection from plant phenotype to indirect selection such as marker-based selection. The innovations in the selection of desirable traits offers optimum resource utilization during cultivar improvement. The major challenges during crop improvement are time and cost during

generation advancement, as it requires growing a large number of selection candidates to select the desirable ones [1,2].

In the last decade, breeding efforts for groundnut (*Arachis hypogaea* L.) have begun to use marker-based early generation selection (MEGS) to reduce costs, enhance the selection-efficiency and shorten the breeding cycle during cultivar improvement [3,4]. In current groundnut MEGS protocols, we use leaf samples for DNA isolation and genotyping, which requires growing of all individual lines carrying desired, as well as undesired, alleles, which requires larger fields or glasshouses. In addition, counting the number of samples for each population, printing and tagging of labels to individual lines/plants and leaf sample collection altogether makes current protocol very time consuming and resource intensive. It takes up to 45 days of time from seed germination to genotypic data generation, data analysis, and interpretation. Therefore, the entire procedure takes significant time and resources in terms of man-power and cost. The seed chipping-based sampling for DNA isolation and genotyping provides an opportunity for early selection of lines with desired alleles, and we can exclude the lines with undesirable alleles from a breeding population before planting comparatively in less time and in an inexpensive way [5]. Recently, single seed chipping-based genotyping (SSCBG) has demonstrated the potential for accelerating breeding programs in few crops [6–8].

Availability of diagnostic markers for foliar diseases (rust and late leaf spot) resistance [9,10] and high oleic acid content [11,12] have provided the opportunity for performing marker-assisted selection (MAS) for improving disease resistance and quality traits in groundnut. As a result, the current groundnut breeding program at ICRISAT shifted from conventional breeding to marker-assisted breeding by deploying the diagnostic markers to perform MAS as well as marker-assisted backcrossing (MABC) to develop high oleic and foliar disease resistant groundnut varieties. These diagnostic markers were recently used to select high oleic acid content and foliar disease resistance traits [13] using leaf-punching at a cost of $2.0–2.5 per sample at Intertek-AgriTech Pvt. Ltd., Hyderabad, India under High-Throughput Genotyping (HTPG) platform [14]. There are now several reports available on deployment of associated markers, i.e., either simple sequence repeats (SSRs)/allele-specific markers or newly developed SNP markers to perform early generation selection in groundnut [15–23]. It is important to note that genotyping through HTPG has increased the adoption of MEGS, MABC, and MAS in groundnut for foliar disease resistance and high oleic acid traits, mainly because of non-requirement of DNA isolation at the breeding institute and faster turnaround of genotyping data for making decisions on selection.

The modified pedigree breeding method, namely, single seed descent (SSD), is regularly used in groundnut breeding programs to advance generations while developing various breeding and mapping populations such as recombinant inbred lines (RIL), nested-association mapping (NAM) and multi-parent advanced generation intercross (MAGIC). The majority of groundnut valuable traits, such as pod yield, shelling outturn, seed size, water stress tolerance, oil and protein content, and disease resistance, are quantitatively inherited with a moderate to low heritability [24]. Selections can be made only in advanced generations ($F_4$ onwards) when the lines accomplish the optimum level of homozygosity. Therefore, rapid generation advancement (RGA) and single seed-based chipping for MEGS can help to reduce the time of each breeding cycle during generation advancement. In this context, in the present study we developed a SSC-based MEGS protocol and compared the selection results of MEGS with the efficiency of phenotypic-based selection in advanced generations. In addition, we analyzed the cost, time, and efficiency of seed chipping and leaf-based methods of DNA isolation and genotyping.

## 2. Materials and Methods

### 2.1. Plant Material and Generation Advancement

We used 24 diverse genotypes (Table S1) for the seed chipping-based sampling for genotyping and germination assessment in the glasshouse. The diverse set of genotypes used in this study included the cultivated *A. hypogaea* subspecies, *hypogaea*, and *fastigiata*,

belonging to Virginia Runner, Spanish Bunch, and Virginia Bunch botanical types. The genotypes were selected based on various traits namely maturity duration, plant habit, foliar disease, fresh seed dormancy, high iron and zinc, high oleic, high yield, and seed features. In order to practice single-seed based genotyping and MEGS, we performed seed chip sampling and selection for >3000 $F_2$ samples for high-oleic trait and foliar disease resistance (Table S2) and hybridity test in 999 $F_1$ samples. Chi-square ($x^2$) analysis was done to test the goodness of fit to the expected segregation patterns for a high oleic trait in $F_2$s.

### 2.2. Sample Collection and DNA Extraction for Genotyping

Leaf tissues of 4 mm diameter were collected using the paper puncher and placed in 12 × 8-well strip tube with a striped cap (Marsh Biomarket, USA) 96 deep well plate (Figure 1). After punching each leaf sample, the paper puncher was cleaned with 70% ethanol to avoid the cross contamination during sample collection. As part of seed-chipping process, a small piece or chip from seed (20 mg/seed) of each genotype was gently cut from the posterior end using an extremely sharp bladed scalpel without disturbing the embryo of the seed and the samples were placed into a 96-well deep plate. In order to keep the identity and track record of seed samples, an acrylic seed-box with the compartment of 96-wells of 28 cm length × 19 cm breadth × 2.5 cm height was designed. The chipped seeds were kept in the same order as that of the seed chip samples stored in a 96-deep-well plate for DNA isolation and genotyping. This seed-box was labelled and stored in a cold-store at 15 °C for 10 days (Figure 1). The DNA was isolated from leaf and seed samples using sbeadex™ (surface-coated superparamagnetic beads; LGC Group, Hoddesdon, UK) along with Kleargene™ (spin columns; LGC Group, Hoddesdon, UK) and the steps were followed according to the manufacturer's instructions at Intertek-AgriTech Pvt. Ltd., Hyderabad, India. Genotyping was performed using the HTPG platform, which facilitates low-cost, fast turnaround data via high-throughput genotyping to research institutes through a collaborative agreement with Intertek-AgriTech (https://www.intertek.com/agriculture/agritech/, accessed on 9 June 2019) as an external service provider [14]. The genotyping data was generated at Intertek-AgriTech using Kompetitive allele specific PCR (KASP) assay for three sets of samples, namely: (a) DNA samples isolated from leaf at ICRISAT using cetyltrimethylammonium bromide (CTAB) method; (b) seed-chipping samples; and (c) leaf-punching samples. In the CTAB DNA extraction method [25], 100 mg of the tender leaf tissue was homogenized in 450 μL of preheated (at 65 °C) extraction buffer and 5 μL mercaptoethanol. The samples were incubated at 65 °C for 1 h and then 450 μL of chloroform-isoamyl alcohol (C:I) (24:1) was added in each sample and mixed thoroughly. The samples were centrifuged at 5500 RPM for 10 min and the aqueous layer (300 μL) was collected in a separate tube. The 210 μL of chilled isopropanol (−20 °C) was added to the collected aqueous layer and mixed thoroughly and centrifuged at 5000 rpm for 10 min. The supernatant was discarded from each sample and the pellet was air dried for 20 min. To remove RNA impurities, the pellet was re-suspended in 200 μL in TE buffer (10 mM Tris EDTA (pH-8)) and then 3 μL RNase (10 mg/mL) was added to each sample, followed by incubation at 37 °C for 30 min. Then, 200 μL of phenol-chloroform-isoamyl alcohol (25:24:1) was added to each sample and inverted twice to ensure proper mixing and the plate was centrifuged at 5000 rpm for 5 min. The aqueous layer was then transferred to fresh tubes and precipitated using 3M sodium acetate (NaOAc), and centrifuged at 5000 rpm for 5 min. The supernatant was discarded and pellets were rinsed with 70% ethanol. Pellets were finally re-suspended in 100 μL low-salt TE and stored at 4 °C [25].

### 2.3. Germination Assay for the Chipped and Control Seed Material

A set of 24 genotypes (chipped) was used for germination assay to study the effect of chipping on germination percentage. We used three controls (intact) and three chipped seeds of each genotype. Replicated planting of genotypes (chipped and control) was done under glasshouse condition. Germination percentage was calculated for each genotype and

compared for chipped and control seeds to study the effect of chipping on seed viability (Figure 2).

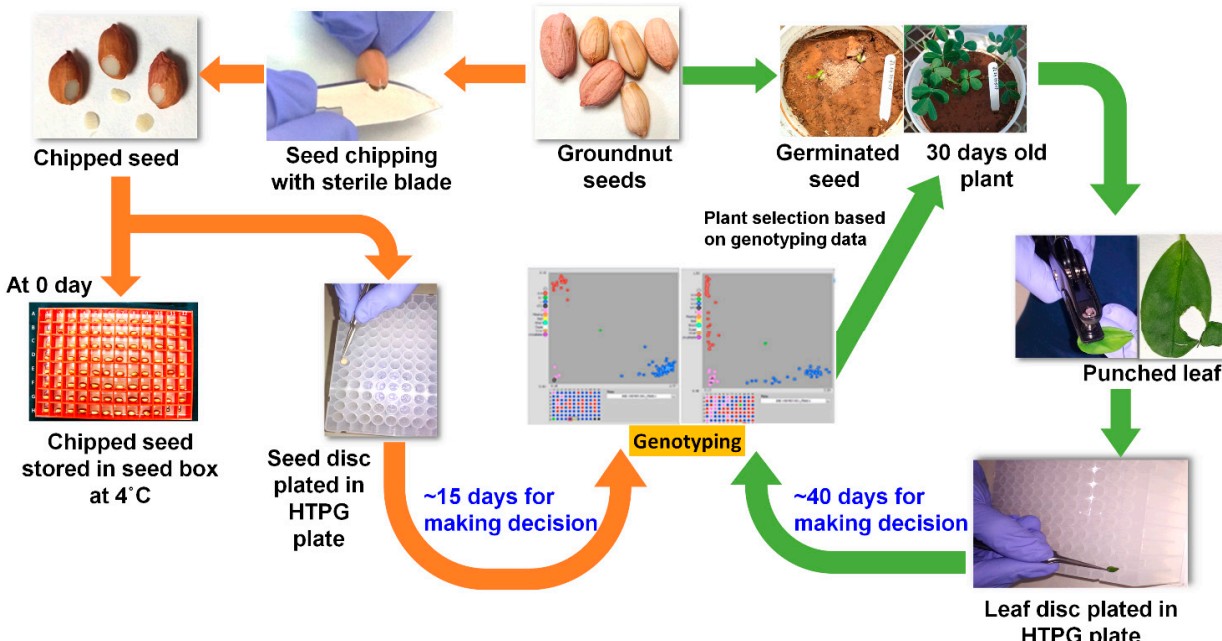

**Figure 1.** Flowchart showing comparison of seed-based genotyping and leaf-based genotyping.

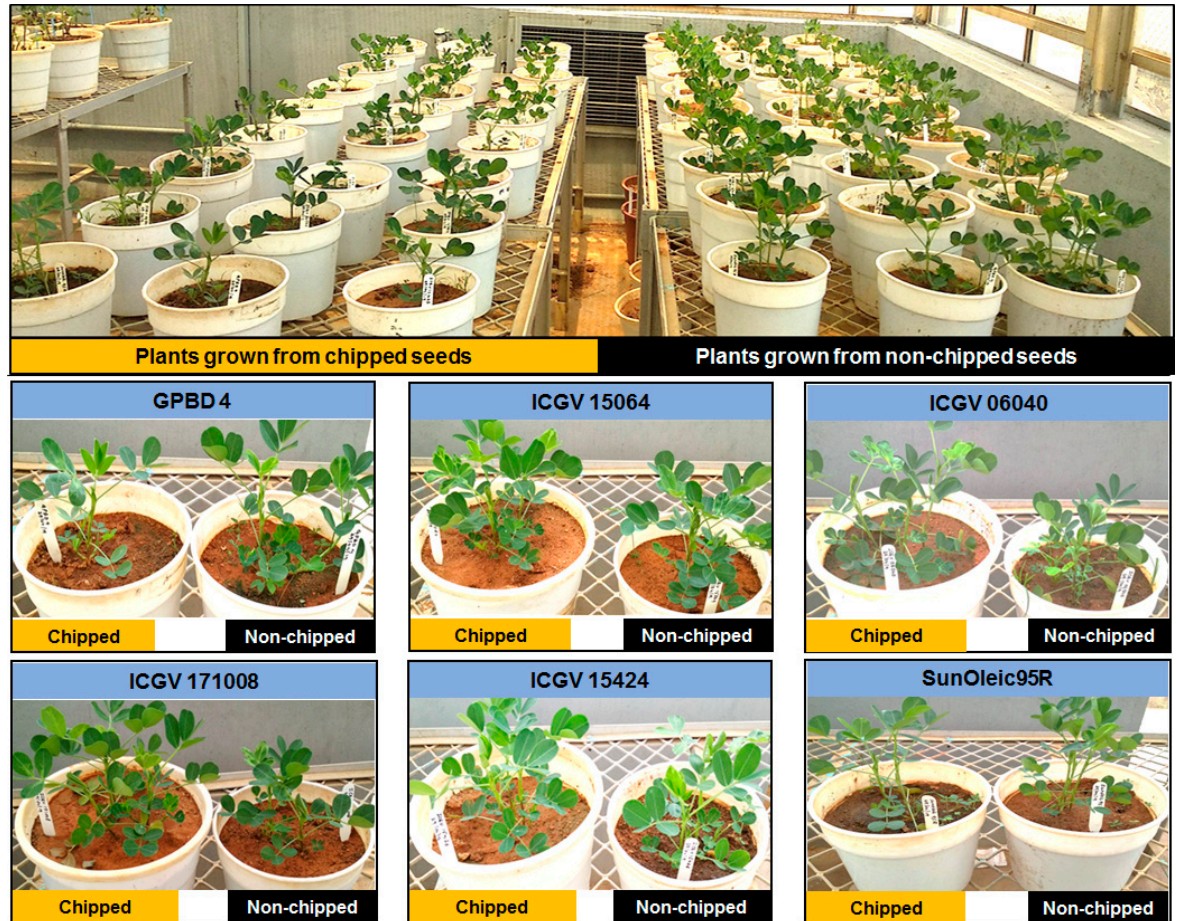

**Figure 2.** Comparative germination rate assessment for chipped and non-chipped samples.

### 2.4. Comparison of DNA Quality and Quantity between Leaf Punching and Seed Chipping Followed by Genotyping Using KASP Markers

A set of 24 genotypes was used to compare the quality (A260/280) and quantity (ng/μL) of DNA isolated from leaf samples and seed samples. The DNA extraction from 24 genotypes was performed at Intertek-Agritech Hyderabad. A set of 24 genotypes were used for genotyping with 10 KASP markers namely snpAH0002, snpAH0004, snpAH0005, snpAH0010, snpAH0011, snpAH0015, snpAH0017, snpAH0018, snpAH0021, and snpAH0026 linked with high oleic acid, rust resistance, and late leaf spot resistance [26] (Figure 3a,b).

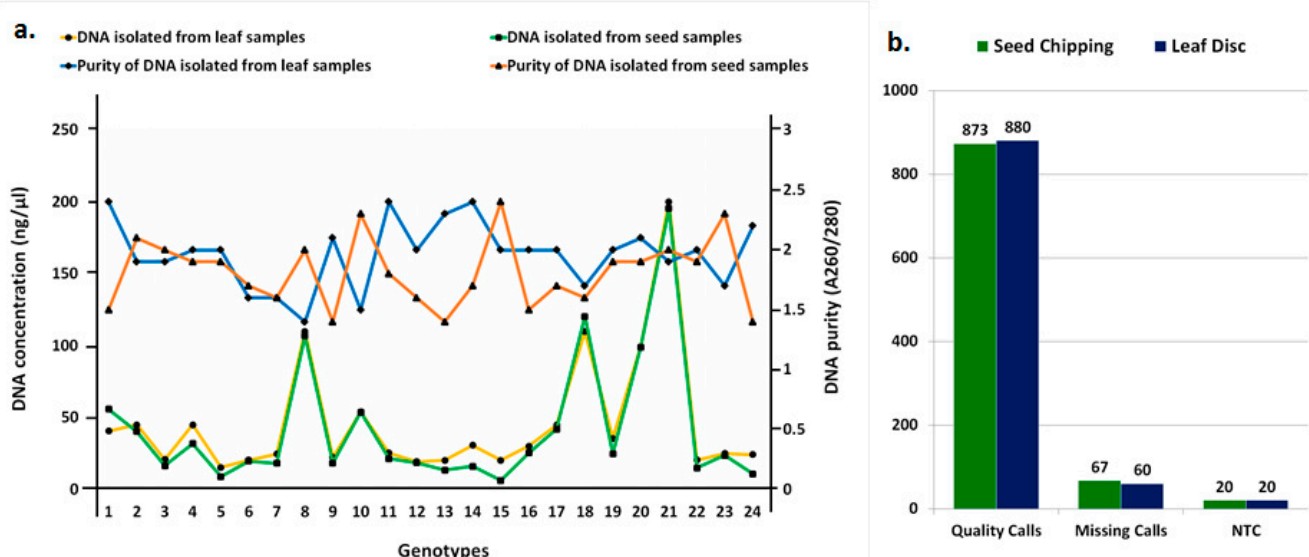

**Figure 3.** Comparative assessment of DNA quantity, quality and quality genotyping calls achieved from leaf disc and seed-chipping samples during genotyping. (**a**) Comparison of DNA quality and quantity between seed chipping and leaf disc samples, (**b**) Comparison of SNP calls between seed chipping and leaf disc samples.

### 2.5. Rapid Generation Advancement House and Crop Growing Facility

In order to optimize the favorable growing environment necessary for RGA, a semi-controlled greenhouse facility of 371 m$^2$ (53 m × 7 m) area was constructed at ICRISAT, Hyderabad (India) (Figure 4). The RGA greenhouse has 12 hollow-cement blocks to facilitate planting. The hollow cement blocks are filled with a soil mixture containing 4 parts of Alfisol, 4 parts of sand, and 1 part of well-decomposed farmyard manure. Steam-sterilization of soil medium at 62 kPa pressure and 82 °C for 1 h. The greenhouse is supported with mist irrigation which helps to create humidity and reduce the temperature by 4 °C compared to an open environment.

The chipping of the seeds did not affect the seed germination percentage even up to 30 days, which is a significant time to complete genotyping and data analysis to select desirable alleles from individuals during generation advancements.

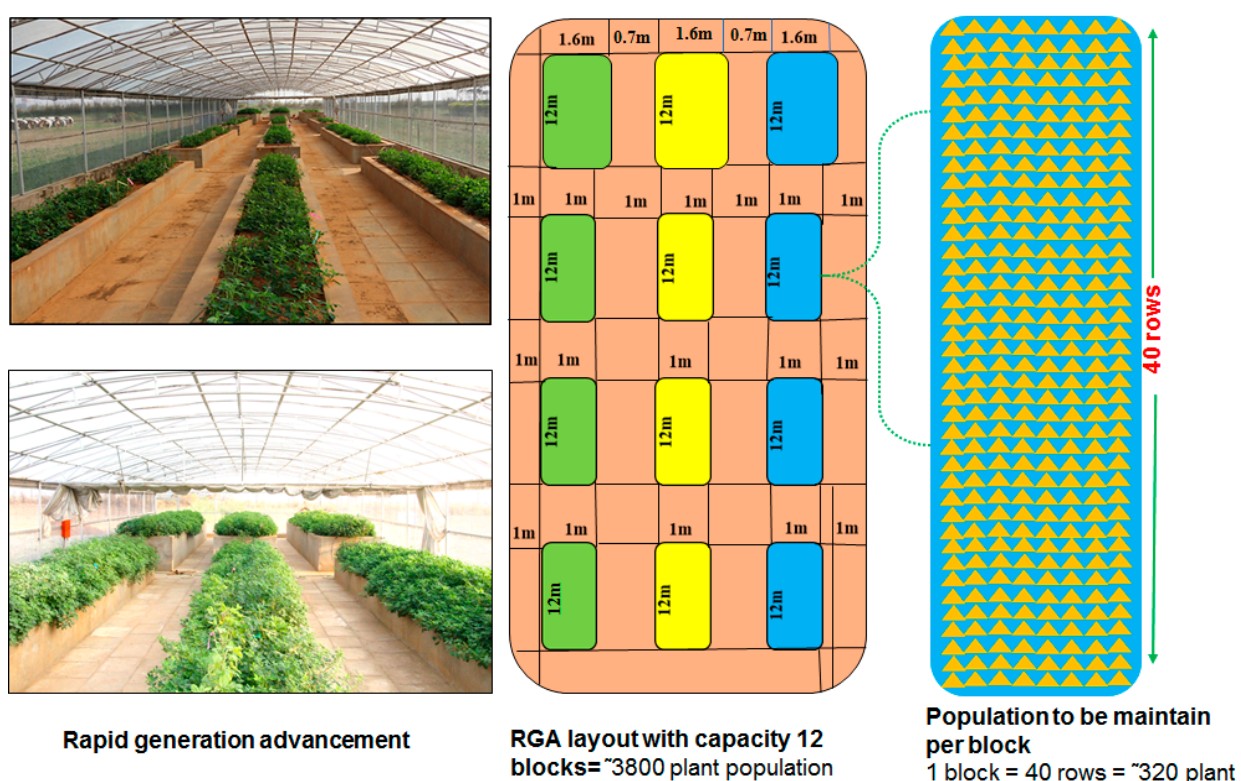

**Figure 4.** Rapid generation advancement (RGA) facility at ICRISAT for taking 3.5 generations per year.

### 2.6. Phenotype-Based Late Generation Selection vs. Marker-Based Early Generation Selection

Selection efficiency was compared between phenotype-based late generation selection (PLGS) following the SSD method vs. marker-based early generation selection (MEGS) in $F_2$ generation. A population derived from ICGV 00348 $\times$ ICGV 15033 cross was advanced individually using SSD and MAS. In phenotype-based selection, true hybrid plants were advanced to $F_2$ and single random seed from each $F_2$ plant was advanced to $F_3$ and subsequently from $F_3$ to $F_4$ generation. In marker-based selection, true hybrid plants were advanced to $F_2$. In $F_2$ generation, the plants were genotyped using FAD alleles and positive plants for *ahFAD2A* and *ahFAD2B* mutant allele were advanced to $F_3$ subsequently $F_3$ plants were advanced to $F_4$ generation. Finally, in $F_4$ generation, the single plant seeds were harvested from both the methods of phenotype-based selection (without markers) and marker based early generation selection were scanned using near infrared reflectance spectroscopy (NIRS) (XDS monochromator, FOSS Analytical AB, Sweden) to select the lines with >75% fatty acid content conferred by *ahFAD2A* and *ahFAD2B* mutant alleles. From the selected plants, the $F_5$ progenies were raised and selections were made based on pod and kernel features, agronomic and plant phenotype. The selection accuracy of each of the selection methods was compared by counting the number of high oleic lines developed from each of the methods.

### 2.7. Cost and Time Comparison among DNA, Leaf-Punching and Seed-Chipping Based Genotyping

The cost and work-force utilized in various operations during leaf disc and seed-chip sampling protocols were compared by deploying both methods on large set of (>3000) lines. The DNA sampling involved the resource costs from glasshouse/field resources for planting, crop husbandry until leaf sampling (~45 days), labelling and tagging, leaf sampling, DNA extraction, and genotyping. The leaf disc sampling involved the resources cost in terms of glasshouse/field resources for planting, crop husbandry until leaf sampling (~40 days), labelling and tagging, leaf-disc sampling, genotyping at Intertek-Agritech, roughing target-allele negative plants, and person-hours of scientists and research technicians needed to accomplish these operations. Seed-chip sampling involved the cost of

resources for seed-box, genotyping at Intertek-Agritech (Figure 5), and person-hours of scientists and research technicians required in seed-chipping and for removing the seed carrying negative allele. The comparative cost of utilized in two protocols is expressed in two different components viz., resources cost and person-hours.

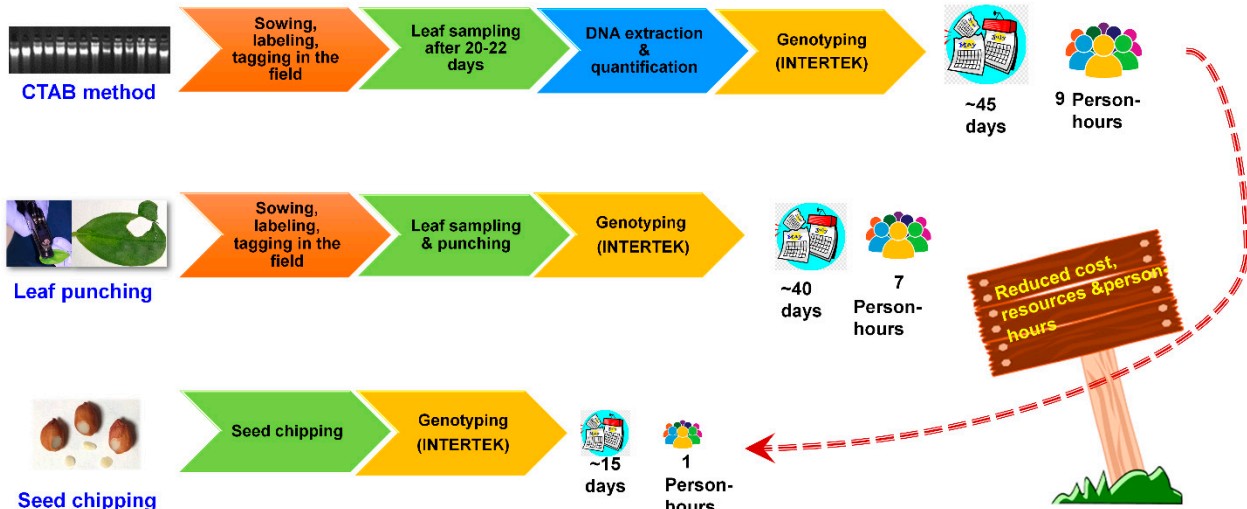

**Figure 5.** Comparative assessment for cost-effectiveness and time saving among DNA-based, leaf-punching, and single seed-chipping based genotyping.

## 3. Results

### 3.1. Comparison of Germination Percentage between Chipped and Non-Chipped (Control) Seeds

In order to assess the effect of seed chipping on germination percentage and seed viability, the chipped and non-chipped (control) seeds of each genotype were planted in the glasshouse with equal agronomic practices and fertilizer dosage. Our results indicated that, both chipped and non-chipped seeds were germinated on the 4th or 5th day after sowing. There was no significant difference in germination percentage for chipped and non-chipped samples (Figure 2). In case of control seeds, the average germination percentage was 83%, and for chipped seeds it was 77% (Table S1).

### 3.2. Comparison of DNA Quality and Quantity between Seed-Chip and Leaf-Disc Samples

We compared the quantity and quality of DNA samples isolated from the seed and leaf tissues of 24 genotypes (Table S1). We observed that the average concentration of DNA from seed was 42.0 ng/μL, however it was higher from leaf at 46.1 ng/μL. The highest concentration of DNA was obtained from seed (195.5 ng/μL) and from the leaf (200.0 ng/μL). In case of quality, a ratio of 260/280 is generally used to determine protein or RNA impurities in isolated DNA. In our findings, we observed that the average of A260/280 ratio from seed was ~1.81, and from leaf it was ~1.96. Therefore, the quality and quantity of isolated DNA from leaf and seed tissues are almost similar. On the basis of these results we concluded that good quality DNA can be isolated from seed as well as leaf (Figure 3a).

### 3.3. Comparison of Genotyping Results between Seed-Chip and Leaf-Disc Samples

Genotyping of leaf and seed DNA samples using 10 KASP markers generated good quality allele calls for each genotype. In the case of chipped samples, for one 96 deep well PCR plate, out of 960 calls (96 samples × 10 SNPs) 873 alleles calls (93%) were generated as high quality calls, while only 67 unknown allele calls (missing) and 20 allele no template controls (NTCs) were generated. In the case of leaf disc samples, a total of 880 allele calls (93.6%), with 60 missing calls and 20 NTCs, were generated. Comparison of genotyping

calls from leaf disc and seed chip indicated that seed chipping samples also generated high quality SNP calls (Figure 3b).

### 3.4. Optimization and Deployment of Seed-Chip Based Genotyping for Early Generation Selection in Groundnut Breeding

The high quality and quantity DNA extraction from seed-chip encouraged us to deploy seed-chip based genotyping in the groundnut breeding populations. A total of 3000 $F_2$ seeds generated from six crosses made for high oleic trait were used for seed-chip sampling followed by DNA extraction and SNP genotyping using KASP marker "*ahFAD2B*" developed for high oleic acid. The SNP-genotyping results for *ahFAD2B* allele generated high quality SNP calls and shown that 593 seeds with homozygous (A:A) alleles, 1327 seeds with heterozygous allele (A:-) and 1028 seeds with wild type allele (-:-) (Table S2). The genotyping data generated was used to calculate the segregation of alleles using Chi-square ($x^2$) test for individual cross. The results showed the expected monogenic inheritance ratios of 1:2:1 for *ahFAD2B* mutant allele fits for three crosses *viz*., ICGX 191003, ICGX 191005, and ICGX 191008, whereas ICGX 191004, ICGX 191006, and ICGX 191,007 showed deviation from the expected monogenic inheritance ratios of 1:2:1.

In addition, we also checked the efficiency of seed-chip sampling to test the hybridity of $F_1$s, which can be used to identify true hybrids ($F_1$s). We sampled 999 $F_1$ seeds from different crosses and extracted the DNA and genotyped using KASP assay. We observed that, of the 999 $F_1$s, a total of 659 hybrid seeds (66%) were confirmed as true hybrid seeds indicating possible increase of 34% efficiency in breeding through marker-based hybridity confirmation. The seeds with homozygous *ahFAD2B* alleles in $F_2$s and heterozygous in $F_1$s as well as $F_2$s were selected for generation advancement and the seeds with wild type *ahFAD2B* allele were discarded by tracking back the sampled seeds stored in the seed-box. The selected seeds were socked in lukewarm water in Petri-dishes and allowed about 72 h for germination. The sprouted seeds were transplanted to an RGA greenhouse facility with narrow spacing ($10 \times 4$ cm) to perform rapid generation advancement (RGA) (Figure 4).

In groundnut, the pegs from a higher node in Spanish and Virginia Bunch types often fail to reach the ground, which leads to the plant producing fewer seeds, especially in $F_2$ where a greater number of seeds is desirable. In $F_1$ nurseries, the plants were generally space planted, and when they attain the flowering stage, the U-shaped clips were attached to branches to hold them close to the surface of soil, which facilitates easy penetration of the pegs into soil. This practice helped in obtaining the desired number of $F_2$ seeds from fewer $F_1$ plants. The seed-chip based sampling for genotyping was followed in the routine breeding pipeline to advance the generations from $F_1$ to $F_3$. After selecting the single plant with desired allelic combination from $F_2$ and $F_3$ population, the single plant selection was performed, and the plants were selected based on pod and kernel features, foliar disease resistance, fresh seed dormancy, fatty acid profile, and oil content. In the $F_5$ generation, progeny bulks were produced and observatory trials were conducted in the $F_7$ generation followed by preliminary and advanced yield trials. Further, currently we can take to 3.5 generations per year in the specially designed semi-controlled-greenhouse facility at ICRISAT. The emphasis is now on scaling up the process and further optimization to accommodate more generations in coming years.

### 3.5. Phenotype Based Selection in Late Generation vs. Marker-Based Selection in Early Generations

This study compared the PLGS and MEGS in groundnut to develop commercially important high oleic groundnut lines (Table S3). In the PLGS (using SSD approach), the $F_4$ population of 1496 plants was developed using the SSD approach, from which the seeds of individual plants were scanned using NIRS and 150 $F_4$ plants were selected with >75% oleic acid content. Of these 150 $F_5$ progenies, the progenies with desirable pod and kernel features, plant growth habit, leaf morphology, basal pegging, and number of primary branches were selected to advance 32 $F_6$ progeny bulks. The 32 $F_6$ progeny bulks were screened in disease nursery for LLS and rust reaction to selected 14 bulks for observation in yield nurseries. In the MAS approach, MEGS was carried out on 1584 $F_2$ plants, and

106 plants were selected that were homozygous for both *ahFAD2A* and *ahFAD2B* mutant alleles to raise106 $F_3$ plant progenies. In the $F_3$ generation, single plant selection was performed based on pod and kernel features and plant growth habit, and in $F_4$ generation selection was performed based on number of primary branches and basal pegging to raise 46 $F_5$ progenies. In the $F_5$ generation, 24 uniform progenies were advanced to the $F_6$ disease screening nursery for rust and LLS reactions. In Rainy 2019, 14 progeny bulks from SSD and 10 from MAS were selected from the disease nursery based on moderate resistance to LLS and rust. The observation nursery selected two lines from PLGS and seven lines from MEGS based on superior pod yield performance in comparison with checks.

### 3.6. Cost and Time Comparison for Genotyping among Different Sample Types (CTAB, Leaf-Punching, and Seed-Chipping)

In the present study, we also estimated and compared the time and cost required for genotyping by three different methods, which were CTAB, leaf punching, and seed chipping. All three methods are entirely different in terms of sample collection, DNA extraction, sample pursuing, and glasshouse management. However, all approaches shared a common genotyping procedure, which is KASP assay-based high throughput genotyping. In case of the CTAB method, the resources cost and time in terms of glasshouse for planting, crop husbandry, leaf sampling, DNA isolation, and genotyping, which takes almost 45 days and has a cost of ~USD760 and 9 person-hours for one 96 deep well plate. Similarly, in the case of the leaf punching method, the resource cost and time in terms of glasshouses for planting, crop husbandry, leaf sampling, and genotyping takes almost 40 days and a cost of ~USD745 and 7 person-hours per plate.

Likewise, for SSCBG it takes 15 days and a cost of ~USD240 and 1 person-hour per plate. This comparison shows that the total cost of seed-chip based genotyping is approximately 3.2 times lower than the CTAB based genotyping and 3 times lower than leaf-based genotyping. In addition, the time of seed-based genotyping is approximately 3 times lesser than the CTAB based genotyping; whereas 2.5 times lesser than the leaf-based genotyping (Figure 5). Therefore, the seed-chip-based genotyping is cost and operationally effective compared to the existing CTAB and leaf punching based DNA extraction methods. Besides, the genotyping time is also shortened by about 25–30 days, as there is no need to raise seedlings to harvest leaf discs.

The person-hours and operational costs involved in leaf disc and seed-chip sampling protocols were carried out at large scale for the 3000 $F_2$ population collected from the field (Table S4). The leaf-based sampling required 194 person-hours for 3000 populations and 9 person-hours per 96 deep well plates involved in the various operations of planting, crop husbandry (for about 45 days till leaf sample collection), label printing and tagging, leaf disc sampling, and roughing target-allele negative plants from the field. In the case of cost, the operational expenses of $8262 and sample collection ~$400 per plate were estimated for 3000 leaf-based samplings. The leaf based sampling process required 45 days for overall operations and genotypic data generation and analysis. On the other hand, seed-chip sampling needed 23 person-hours for 3000 populations and 7 person-hours per plate involved in seed-chip sampling. In the case of cost, the operational expenses of $8064, and for sample collection ~$240 per 96 deep well plate, were estimated for 3000 seed-chip and 15 days for the genotypic data generation and analysis (Figure 5). Thereby, our findings indicated the seed-based strategy is more convenient, reliable, accurate, and inexpensive compared to both CTAB and leaf punching method; suggesting the seed-based method is more useful for accelerated breeding programs.

## 4. Discussion

Reference genomes of both subspecies of cultivated groundnut are available for use in different genomics and breeding applications [27–29]. Extensive efforts have been implemented over the last decade for trait dissection and the use of breeding populations and discovery of associated genomic regions, candidate genes, and diagnostic markers in groundnut [1,4,30,31]. Currently, diagnostic markers are available in groundnut breeding

for resistance to foliar fungal diseases, nematode, bacterial wilt, and high oleic acid, seed weight and fresh seed dormancy traits. Marker-assisted selection (MAS) approaches for different traits such as, nematode resistance [32], high oleic [11,13,19,23] and rust, and LLS [16–18] using diagnostic markers of various marker systems such as SSRs, ASP (allele-specific), AhMITE1 (*A. hypogaea* miniature inverted-repeat transposable element), and SNP markers are used by the groundnut breeding programs in India, the USA, and China. Through a cooperation agreement with Intertek-AgriTech, the HTPG project, directed by ICRISAT, intends to deliver low-cost, high-throughput genotyping services primarily for forward breeding applications [14]. Deployment of a combination of SNP markers for pyramiding different traits provides cost-effectiveness by deploying MEGS to select desired lines [13,23]. MEGS reduces the population size manageable with limited resources.

DNA isolation for a large population from leaf tissue is a challenging task for a breeding program. In the DNA isolation process, resources and person-hours are the major issues. Leaf-disc based genotyping involves several operational steps, such as planting of $F_2$ populations in fields/glasshouses, counting the plant stand per population, label printing and tagging of each $F_2$ plant, collection of the leaf discs from each $F_2$ plant, coming again to the field with genotype data, scanning the labels to select the desirable plants, and rejecting undesirable ones. All of these operations generate additional financial burdens as they involve a significant number of hours of person-power and operational costs. In addition, the process is also prone to human errors, as it involves several steps. In contrast, the seed-based genotyping approach provides an alternative approach in cost-effectiveness and time-saving strategies for accelerated breeding programs. The DNA isolation from seed and its PCR compatibility has been already shown in previous studies in few important crop plants [5–7,33]. In maize, SSC-based DNA isolation and genotyping was also attempted during marker-assisted selection for achieving higher genetic gains [7]. In addition, it was also used for MAS in rice and soybean [34].

In the present study, we have demonstrated an integrated approach SSCBG, for MEGS of desired lines during generation advancement in a groundnut breeding program. In this method, we used a 20 mg seed chip from posterior end of a single seed cotyledon for DNA extraction. A sharp bladed scalpel is highly recommended, so that a small seed chip from a hard seed can be easily sliced off without damaging the embryo. As a result, the seed material in the form of seed chip yielded adequate and high-quality DNA for high-throughput genotyping using 10 SNP panels with a high quality allele call rates for 10 SNPs. The quality and quantity of DNA from the chipped seed samples were compared with the DNA obtained from the punched leaf samples. During quality analysis using a Nanodrop a ratio of 260/280 for DNA isolated from seed was ~1.81 and from leaf it was ~1.96, which indicated that even seed material can also yield high quality DNA. There are several reports on seed-based DNA used for PCR based genotyping to demonstrate their importance in breeding programs [5–8,35–38].

Unlike monocots, the groundnut seed is susceptible to the splitting of cotyledons due to its dicotyledonous nature, which can affect seed germination. Through the present paper, we report that seed chipping does not affect seed germination. Because the chipped and non-chipped seeds of genotypes used in this study showed a similar germination percentage even after 30 days of storage at 4 °C. As seed-chipping-based genotyping does not require prior seed germination or their planting in the field for identification of true $F_{1s}$; this method could be explored for various purposes in breeding programs, such as the selection of desirable line in large segregation population, identification of duplicates in germplasm, estimation of genetic purity of available varieties, and confirmation of hybrids derived from two parent or multi-parent crosses. Seed-chipping-based genotyping can be carried out in batch mode unless a minimum target number of desirable genotypes is found. It provides an advantage of selection of a minimum number of desired genotypes/lines without prior growing of plants in the field/greenhouse; which reduces the unnecessary expenses required during maintenance of undesired and desired lines/accessions in the field during leaf-based genotyping methods. The segregation in 3000 $F_{2s}$ derived from

six crosses showed a goodness of fit to the ratio of 1:2:1 for *ahFAD2B* allele in $F_2$ seeds obtained from three crosses using SSD approach of genotyping (Table S2). In some crosses we also observed the deviation of chi-square ($x^2$) test from the ratio 1:2:1. The deviation from the expected monogenic segregation ratio in some of the crosses could be due to the sub-sampling $F_2$s or the presence of genetic modifiers [39]. Further, the line development using SSD approach involving NIRS based selection for high oleic trait in $F_4$ generation was compared with MEGS in $F_2$. NIRS scanning was performed based on the calibrations ($R^2 = 0.9$) developed on bulk seed, which is more cost-effective than selection using SNP genotyping. The scanning of 1500 samples can be completed by two persons in six days. In the present study, the SSD approach selected the 14 best bulks, while the MAS selected the 10 best bulks. The MEGS has the ease of managing the populations by the breeding program as the population size is narrowed down in $F_2$ generation as we are dropping undesired lines. In contrast, in order to advance a large population till $F_4$ by using SSD approach to apply PLGS requires additional resources and sometimes unmanageable by the breeding units when SSD for 70 crosses are to be advanced from $F_2$ to $F_4$ generation. The recovery of desirable homozygotes in the advanced generation ($F_6$) was higher than in the $F_2$ generation. In case of FAD mutant alleles, the both loci are independent representing two groundnut sub-genomes. Therefore, for FAD alleles the frequency of homozygotes in $F_2$ was 0.0625 which was increased to 0.235 in $F_6$ generation. Therefore, it is possible to maintain a population of 450 to select 100 plants in $F_6$ instead of maintaining large population 1584 used in the study to select 106 in $F_2$. However, advancing selections in $F_6$ requires rapid generation turn over, where the fixation can be cost-effectively achieved in a short period of time, such as that of rice [6]. With the current turnover of three cycles per year and with the available capacity, early generation selection in $F_2$ will cut down the population from 1000′ s to 100′ s and the selection can be carried out on pod and kernel in $F_3$ generation. The $F_1$ to $F_4$ are carried forward in the available cost-effective rapid turnover and the $F_4$ progenies are planted in breeding nurseries.

Most importantly, cost and time are important parameters for research program activities. In the present paper, we discussed three methods for genotyping such as, CTAB, leaf-punching, and seed-chipping. The single seed-based genotyping method was the fastest and could deliver the results in 15 days in an inexpensive way (250USD/plate). This cost is substantially less than for the CTAB or leaf disc-based methods, which require the expenditure of cost around for glasshouse 760 USD/plate and could require time up to 45 days.

Further, for a population size of 3000, the seed-chip based method reduced the operational expenses by 3%, work-force by eight folds and generated the genotypic data by 15 days as compare to leaf disc-based method. Therefore, utilization of seed-based genotyping for early generation selection of desired alleles in a breeding population helps to enhance the selection efficiency and genetic gains. Seed-chip based sampling can be deployed in different HTPG programs, which utilize mid (5K SNPs) or high-density SNPs panel (50K SNPs). Therefore, the integration of SSCBG and RGA holds great promise in increasing the selection efficiency as well as accelerating the process of genetic/breeding population development in groundnut (Figure 6).

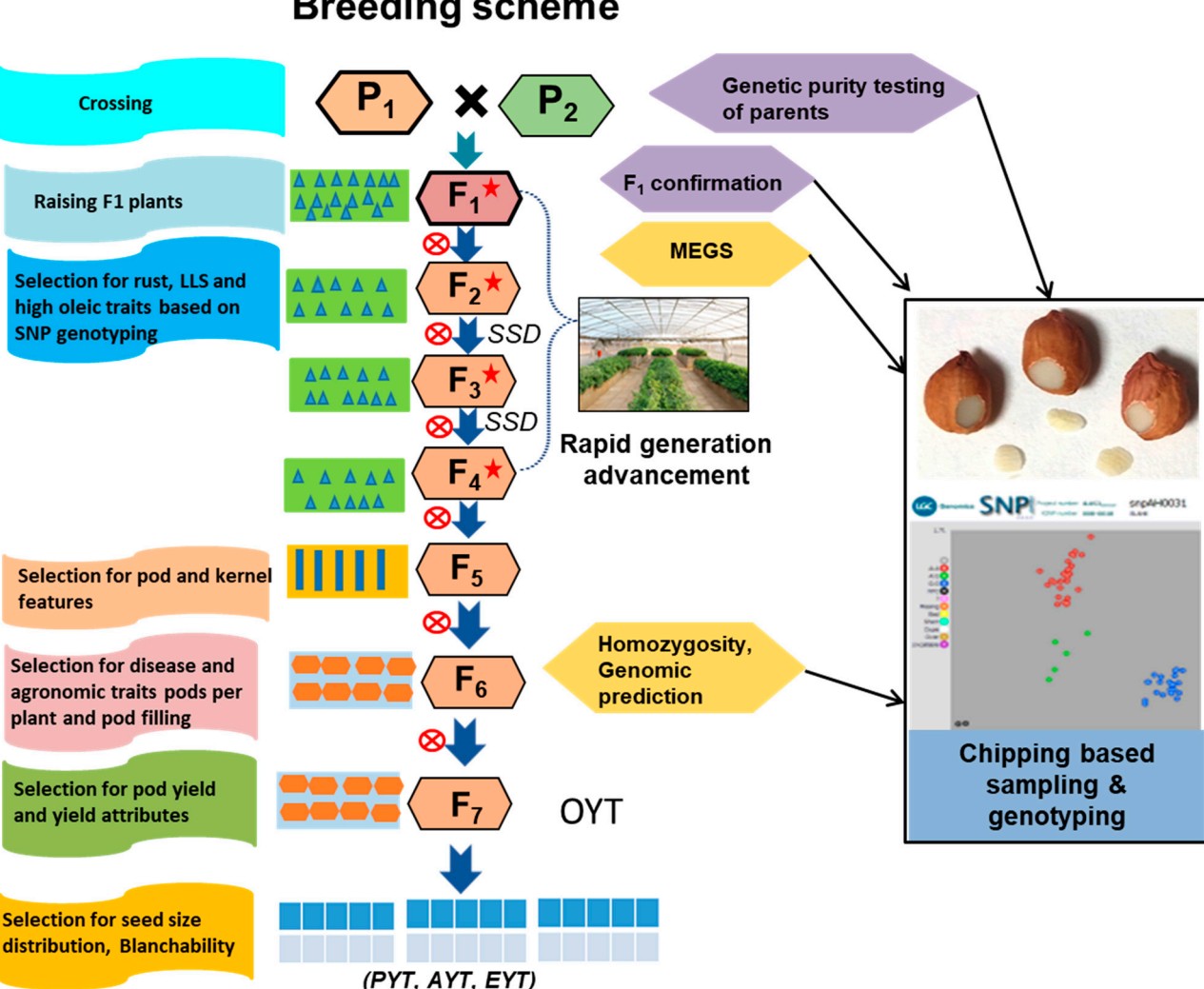

**Figure 6.** Applications of seed-chipping in routine groundnut breeding program and its integration with rapid generation advancement for accelerated groundnut breeding.

## 5. Conclusions

The HTPG project provided an option of genotyping 10 SNPs for just 2.0–2.50 USD per sample for performing early generation selection in several crops, including groundnut. Leaf disc samples were replaced with seed-chip for HTPG in the ICRISAT groundnut breeding program for cost-effectiveness and increased operational efficiency. The study presents the deployment of single-seed based genotyping in breeding for high oleic groundnut together with innovations like storing and treating the chipped seeds to obtain >90% germination, and increased number of $F_2$s from fewer $F_1$ plants. The segregation in 3000 $F_2$'s derived from six crosses show a goodness of fit to the ratio of 1:2:1 for *ahFAD2B* allele in $F_2$ seeds obtained from three crosses using single seed approach of genotyping (Table S2). Deviation from the expected monogenic segregation ratio is attributed to the sub-sampling, contamination, or the presence of genetic modifiers.

Integration of the low-cost genotyping platform with seed-chipping further lowers the cost of genotyping. This study suggests that it is advantageous to replace DNA (isolated in

the lab) and leaf-based genotyping with SSCBG in groundnut to reduce time and resources needed. Further, comparison of the SSD approach with marker-assisted selection in an early generation for breeding high oleic groundnut lines indicates the possibility of combing these to make the selection of advance generation fixed lines. However, cost-effectiveness for such an approach can be achieved when rapid generation tools that can advance 5–6 generations per year are available. For the groundnut breeding program at ICRISAT, the low-cost rapid generation turn-over facility can handle three generations per year. Thus, $F_1$ to $F_4$ are handled in this facility and the $F_4$ selected plant progenies are planted in the field for selection of agronomic traits. The early generation marker-assisted selection approach based on the seed-chip genotyping reduced the $F_2$ to ca. 100 $F_2$ seeds that were planted in the facility as against advancing ca. 1500 $F_2$s through SSD. The low-cost generation advancement facility also enables selection of $F_3$ plants based on pod and kernel features. The application of seed-chipping technology for early generation is complemented by the low-cost rapid generation advancement for cost-effective development of high oleic groundnut line breeding and cultivar improvement in groundnut with an enhanced rate of genetic gain.

**Supplementary Materials:** The following are available online at https://www.mdpi.com/article/10.3390/agronomy11061226/s1, Table S1: Summary of DNA quality and quantity between leaf-disc and seed-chipped, and germination rate between chipped and non-chipped seed samples among 24 genotypes, Table S2: Seed-chip based genotyping results of F2 population for *ahFAD2B* allele, Table S3: Generation advancement in a cross (ICGV 00348 × ICGV 15033) using phenotyping-based section vs marker-based selection, Table S4: Cost comparison for leaf-punching and seed-chipping based sampling and genotyping in 3000 breeding population at ICRISAT-Patancheru, India.

**Author Contributions:** M.K.P. conceived the idea and coordinated experiments on comparison of DNA, leaf-punching and single seed-chipping based genotyping, germination assessment under glass-house conditions and cost estimates. P.J. (Pasupuleti Janila) conceived the idea and coordinated experiments integrating use of seed-chipping and rapid generation advancement in breeding. S.P., D.B.D., S.C., M.T.V., S.S.M., V.S., and P.J. (Pushpesh Joshi) performed the experiments, summarized results and S.P. and D.B.D. drafted the manuscript. R.K., P.J. (Pushpesh Joshi), S.S.M. and V.S. contributed in glasshouse, field and low-cost rapid generation experimentations. R.B., P.S., S.S.G. and R.K.V. contributed in genotyping samples and improving draft manuscript. All the authors read and approved the content of the Manuscript. All authors have read and agreed to the published version of the manuscript.

**Funding:** This research was partially funded by the National Agricultural Science Fund (NASF) of Indian Council of Agricultural Research (ICAR), Department of Biotechnology (DBT), India and Bill & Melinda Gates Foundation (High Throughput Genotyping Project—HTPG (OPP1130244), USA, and OPEC Fund for International Development (OFID)).

**Institutional Review Board Statement:** Not applicable.

**Informed Consent Statement:** Not applicable.

**Data Availability Statement:** Data made available in Supplementary Files.

**Acknowledgments:** The authors are thankful for financial support from Department of Biotechnology (DBT), India; Bill & Melinda Gates Foundation (Tropical Legumes III, High Throughput Genotyping Project—HTPG, USA, and OPEC Fund for International Development (OFID). SP acknowledges Council of Scientific and Industrial Research (CSIR), Govt. of India for the award of Junior Research Fellowship for PhD research. This work has been undertaken as part of the CGIAR Research Program on Grain Legumes and Dryland Cereals (GLDC). ICRISAT is a member of CGIAR Consortium.

**Conflicts of Interest:** The authors declare that they have no conflict of interest.

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
