# Peer review of "Single Seed-Based High-Throughput Genotyping and Rapid Generation Advancement for Accelerated Groundnut Genetics and Breeding Research"

_agronomy, doi:10.3390/agronomy11061226_

Round 1
Reviewer 1 Report
The manuscript describes a method on single seed-based DNA extraction and genotyping for groundnut. It is straightforward. The study is of significance for accelerating breeding. Please see the following comments and suggestions for changes:
- There are lots of acronyms. Please make sure that an acronym appears in the text and are used afterwards.
- Acronym HTPG, High-Throughput Genotyping (page 2) and High-Throughput Phenotyping and Genotyping (page 3): Which one is correct? Please keep consistent.
- This study focuses on the comparison of seed-chipping and leaf punching samples. Why was CTAB method included and considered as a set of samples?
- Results 3.2 Comparison of genotyping results between seed-chip and leaf-disc samples: There are no meaningful results included in this part.
Author Response
Response to the comments from reviewer #1
Comment 1: There are lots of acronyms. Please make sure that an acronym appears in the text and are used afterwards.
Authors’ Response: Thank you for this important suggestion. Now we have elaborated all the acronyms used in revised version.
Comment 2: Acronym HTPG, High-Throughput Genotyping (page 2) and High-Throughput Phenotyping and Genotyping (page 3): Which one is correct? Please keep consistent.
Authors’ Response: Thank you for bringing this typo error to our notice. The correct full form of HTPG is “High throughput genotyping”. Now, we have corrected the HTPG full form on page 3 as “High throughput genotyping.
Comment 3: This study focuses on the comparison of seed-chipping and leaf punching samples. Why was CTAB method included and considered as a set of samples?
Authors’ Response: Thank you for raising this important point. CTAB DNA extraction method is reliable and inexpensive for plant DNA isolation. Most of the laboratories follow CTAB DNA extraction protocol and use for performing genotyping. Therefore, one of the objectives of present study was to compare the DNA quality between leaf discs extracted using CTAB method in lab as well as the DNA quality extracted from leaf discs and chipped seed by the service provider, i.e., Intertek. The inclusion of CTAB based DNA isolation helped us in comparing the cost with seed chipping and leaf disc based sampling approaches.
Comment 4: Results 3.2 Comparison of genotyping results between seed-chip and leaf-disc samples: There are no meaningful results included in this part
Authors’ Response: To provide more clarity on the comparison of genotyping calls from leaf disc and seed chipping, we have included the graph (Figure 3a) in the manuscript. In addition, we have provided detailed explanation on comparison of quality of SNP calls generated from seed chipping and leaf disc. Following graph is added in main manuscript (Figure 3a).

Reviewer 2 Report
In this Research Article, Parnar et al report the application of a seed-based high throughput genotyping approach that they combined with a controlled plant growth cultivation system to reduce the selection time between plant generations. They then combine and implement these techniques for a groundnut breeding program. I think that the approach conducted and the data shown are convincing and a good fit for publication in Agronomy. The following comment need to be tackled to improve the quality of the report:
Major:
1) Since DNA isolation is a key aspect of the method, it would be important to have a more detailed description of the method used for measuring DNA concentration and quality. Furthermore, it would be good that the author further describe the CTAB method used for DNA extraction.
2) I agree with the authors that genotyping F2 seeds and then transferring selected plants saves personnel work for their breeding program. However, I think that the way the method are compared in Figure 5 overestimates the advantage of the seed chipping method. Indeed, although smaller than the full population, selected plants will still need to be transferred to soil for propagation. Furthermore, with this DNA isolation method, the time between plant generations is not shorter as the selected plants need to be propagated in the same way and thus does not accelerate breeding. Figure 5 needs to be modified to take this into account. This should also be discussed in the Discussion section. I think that it would actually be more insightful to calculate the difference of cost and personnel time for the whole breeding program using different techniques or between F1 and F3 instead of just analyzing the the genotyping process.
3) In figure 2, it would be good to have the comparison of plant growth for all the genotypes analyzed (24 genotypes). Those could also be put as supplemental data. It seems from the first panel that chipped plants are smaller than non-chipped one. This seems not to be the case when the 6 individual genotypes are shown.
4) In figure 3, for the comparison of DNA concentration, all measurements could be combined and a statistical test could be run to compare the measurement of the 2 DNA isolation methods. Showing variation between individual technical replicates would also improve the quality of the data.
5) In figure 6 it would be good to indicate at which step of the breeding program was the genotyping done.
Minor:
- Change arrows and text box positions in figure 1 to make the figure it clearer, some of the elements are currently overlapping.
- Figure 4: change blue box size and yellow triangle alignments on schematic.
- It is mentioned in the conclusion that more than 90% of seeds germinate, what was the total number of plants analyzed?
- In the introduction (for RIL, NAM, MAGIC) put first full name of the technique and then abbreviation in parenthesis instead of the opposite. Add full name for RGA
Author Response
Response to the comments from reviewer #2
Major comments
Comment 1: Since DNA isolation is a key aspect of the method, it would be important to have a more detailed description of the method used for measuring DNA concentration and quality. Furthermore, it would be good that the author further describe the CTAB method used for DNA extraction.
Authors’ Response: Thank you for this important suggestion. Now we have provided a detail CTAB DNA extraction protocol in the main manuscript. Following protocol have been provided,
In CTAB DNA extraction method, 100 mg tender leaf tissue homogenized in 450 µl of preheated at (65 ºC) extraction buffer with 5 µl mercaptoethanol added. The samples were incubated at 65 ºC for 1 hr and 450 µl of chloroform-isoamyl alcohol (C:I) (24:1) was added in each sample and mixed thoroughly by inverting. The samples were centrifuged at 5,500 RPM for 10 min and the aqueous layer (300 µl) was collected and 210 µl of chilled isopropanol (–20 ºC) was added to the collected aqueous layer and mixed thoroughly by inverting and centrifuged at 5,000 rpm for 10 min. The supernatant was discarded from each sample and the pellet was air dried for 20 min. To remove RNA impurities, the pellet was re-suspended in 200 µl in TE buffer (10 mM Tris EDTA (pH-8)) and 3 µl RNase (10 mg/ml) was added to each sample and incubated at 37 ºC for 30 min. 200 µl of phenol-chloroform-isoamyl alcohol (25:24:1) was added to each sample and inverted twice to ensure proper mixing and the plate was centrifuged at 5,000 rpm for 5 min and centrifuged at 5,000 rpm for 5 min. The aqueous layer was transferred to fresh tubes and precipitated using 3M sodium acetate (NaOAc) and centrifuged at 5,000 rpm for 5 min. The supernatant was discarded and pellet was washed with 70% ethanol. Pellet was re-suspended in 100 µl low-salt TE and stored at 4 ºC.
Comment 2: I agree with the authors that genotyping F2 seeds and then transferring selected plants saves personnel work for their breeding program. However, I think that the way the method are compared in Figure 5 overestimates the advantage of the seed chipping method. Indeed, although smaller than the full population, selected plants will still need to be transferred to soil for propagation. Furthermore, with this DNA isolation method, the time between plant generations is not shorter as the selected plants need to be propagated in the same way and thus does not accelerate breeding. Figure 5 needs to be modified to take this into account. This should also be discussed in the Discussion section. I think that it would actually be more insightful to calculate the difference of cost and personnel time for the whole breeding program using different techniques or between F1 and F3 instead of just analysing the genotyping process.
Authors’ Response: We agree that the time between the generations will not be shortened by seed chipping method. It saves resources and saves personal time and work to a great extent and that same is presented in Figure 5. The Figure 5 only compares only the genotyping time, personal time and describes operations involved in three different methods and not the generation time. To avoid confusion between breeding generation time with the genotyping time as sentence is added in the discussion section.
Comment 3: In figure 2, it would be good to have the comparison of plant growth for all the genotypes analyzed (24 genotypes). Those could also be put as supplemental data. It seems from the first panel that chipped plants are smaller than non-chipped one. This seems not to be the case when the 6 individual genotypes are shown.
Authors’ Response: Thank you for this important suggestion, however, we have taken observation on germination percentage only and not the other morphological features. Nevertheless, we have noted this important suggestion and will be include in our future such studies to gain more insights.
Comment 4: In figure 3, for the comparison of DNA concentration, all measurements could be combined and a statistical test could be run to compare the measurement of the 2 DNA isolation methods. Showing variation between individual technical replicates would also improve the quality of the data.
Authors’ Response: Thank you for this important suggestion. We have included the statistical parameters such as Standard error, standard deviation, minimum, maximum and mean for purity, concentration as well as the germination percentage in Supplementary Table 1. In Figure 3, we have provided the graph for comparison of DNA quality and purity between seed chips and leaf discs.
Comment 5: In figure 6 it would be good to indicate at which step of the breeding program was the genotyping done.
Authors’ Response: Thanks for this important suggestion. In figure 6, we have shown arrows at F1, F2 and F6 step of the breeding program where we can do the seed chipping for selection to save time of a breeder.
Minor comments
Comment 1: Change arrows and text box positions in figure 1 to make the figure it clearer, some of the elements are currently overlapping.
Authors’ Response: Authors thank the Reviewer#2 for this important suggestion. We have now changed the arrows in Figure 1.
Figure 1. Flowchart showing seed-based genotyping and leaf-based genotyping.
Comment 2: Figure 4: change blue box size and yellow triangle alignments on schematic.
Authors’ Response: We have now updated the figure 4 in the revised version of MS.
Comment 3: It is mentioned in the conclusion that more than 90% of seeds germinate, what was the total number of plants analyzed?
Authors’ Response: Authors understand the concern raised. We took 2 replications of 24 genotypes means 48 (24 × 2) plants for chipped and 48 (24 × 2) plants for non-chipped. Percentage of germination was calculated based on 24 genotypes in each replication.
Comment 4: In the introduction (for RIL, NAM, MAGIC) put first full name of the technique and then abbreviation in parenthesis instead of the opposite. Add full name for RGA
Authors’ Response: Authors thanks for the important suggestion. We have provided full forms for all the techniques in present study.
